# *Andrographis paniculata* Extract Supports Skin Homeostasis by Enhancing Epidermal Stem Cell Function and Reinforcing Their Extracellular Niche

**DOI:** 10.3390/cells14151176

**Published:** 2025-07-30

**Authors:** Roberta Lotti, Laetitia Cattuzzato, Xuefeng Huang, David Garandeau, Elisabetta Palazzo, Marika Quadri, Cécile Delluc, Eddy Magdeleine, Xiaojing Li, Mathilde Frechet, Alessandra Marconi

**Affiliations:** 1DermoLAB, Surgical, Medical, Dental and Morphological Sciences Department, University of Modena and Reggio Emilia, 41124 Modena, Italy; elisabetta.palazzo@unimore.it (E.P.); marika.quadri@unimore.it (M.Q.); 2Research and Development Department, Lucas Meyer Cosmetics, Clariant, 31036 Toulouse, France; laetitia.cattuzzato@lucasmeyercosmetics.com (L.C.); david1.garandeau@lucasmeyercosmetics.com (D.G.); cecile.delluc@lucasmeyercosmetics.com (C.D.); eddy.magdeleine@lucasmeyercosmetics.com (E.M.); mathilde.frechet@clariant.com (M.F.); 3PROYA Cosmetics Co., Ltd., Hangzhou 310023, China; huangxuefeng70@proya.com (X.H.); lixiaojing@proya.com (X.L.)

**Keywords:** *Andrographis paniculata*, skin homeostasis, stem cells, dermo-epidermal junction

## Abstract

Skin aging is characterized by compromised epidermal homeostasis and dermo-epidermal junction (DEJ) integrity, resulting in reduced stem cell potential and impaired tissue regeneration. This study investigated the effects of *Andrographis paniculata* extract (APE) on keratinocyte stem cells (KSCs) and DEJ composition in human skin. Using human skin explants and cell culture models, we demonstrated that APE treatment enhances DEJ composition by increasing Collagen IV and Laminin production while decreasing MMP-9 expression, without altering epidermal structure or differentiation. In the same model, APE preserved stemness potential by upregulating markers related to niche components (collagen XVII and β1-integrin), proliferation (Ki-67 and KRT15), and stem cell capacity (Survivin and LRIG1). In vitro studies revealed that APE selectively stimulated KSC proliferation without affecting transit amplifying cells and promoted Collagen IV and Laminin secretion, particularly in KSCs. Furthermore, in a co-culture model simulating a compromised DEJ (UVB-induced), APE increased Laminin production in KSCs, suggesting a protective effect against photo-damage. These findings indicate that APE enhances DEJ composition and preserves stem cell potential, highlighting its promise as a candidate for skin anti-aging strategies targeting stem cell maintenance and extracellular matrix stability to promote skin regeneration and repair.

## 1. Introduction

Epidermal homeostasis is maintained by stem cells in the basal layer, which support continuous skin renewal [1,2]. Three main keratinocyte stem cell (KSC) pools exist: in the interfollicular epidermis, hair follicle bulge, and sebaceous glands [1]. Two models describe stem cell regulation: the hierarchical model, where slow-dividing stem cells produce transient amplifying (TA) cells that differentiate, and the stochastic model, where progenitors randomly self-renew or differentiate [3,4]. Both models likely coexist in the skin. The molecular and cellular signals that govern specific cell fate choices involve the extracellular matrix (ECM), intrinsic cellular signaling pathways, as well as regulation by hormones and adjacent stromal cells [5,6].

Interfollicular epidermal stem cells reside in specialized niches at the dermo-epidermal junction (DEJ), where they receive crucial support from matrices and soluble factors through ligand–receptor interaction [7]. The key pathways involved in maintaining stemness, regulating self-renewal, proliferation, and differentiation, include Wnt/β-catenin, Notch, epidermal growth factor (EGF) receptor, and integrin signaling [8,9]. Specifically, β_1_-integrin, a receptor for collagen found on both keratinocytes and dermal fibroblasts, recognizes specific sequences within collagen, such as GFOGER [10]. This recognition triggers the autophosphorylation of focal adhesion kinase (FAK) and activates downstream signaling pathways, including Akt and extracellular signal-regulated kinase (ERK), which in turn stimulate cellular proliferation and migration [11]. High β_1_-integrin expression has been used to isolate KSC from TA cells [12] and is associated with multipotent differentiation potential in vitro [13].

Skin aging is marked by a gradual reduction in regenerative capacity, primarily due to the functional exhaustion of epidermal stem cells [14]. Among the key markers associated with these cells, LRIG1 [15], Survivin [16,17], and Keratin 15 (KRT15) [18] play crucial roles in maintaining epidermal homeostasis, proliferation, and self-renewal. During aging, the expression levels of LRIG1, Survivin, and KRT15 are significantly reduced, reflecting a loss of stemness and diminished repair potential [19,20,21]. In addition to the reduced proliferative capacity of stem cells, aging also leads to changes in the surrounding microenvironment [22]. Key components of the DEJ, including laminins and collagens, undergo structural and functional alterations that impair stem cell activity [23]. Age-related changes and environmental stress further disrupt the balance between protein synthesis and degradation [24]. Matrix Metalloproteinases (MMPs) become overexpressed and more active, leading not only to the degradation of collagen and elastin fibers in the dermis but also to the breakdown of DEJ components. This disruption alters the mechanical properties of the DEJ and influences cellular behavior by modifying the stiffness of the microenvironment [25]. These age-related modifications in both stem cells and their niche contribute to reduced regenerative potential and tissue maintenance, which are hallmarks of aging [26].

Exposure to UV light triggers the production and activation of MMP-1, -3, and -9, leading to the breakdown of laminins, collagens, nidogen, and perlecan. In naturally aging skin, senescent cells contribute to MMP production, promoting degradation of the dermal–epidermal junction. This leads to structural damage, which is even more pronounced in photo-aged skin [27]. Aging causes the skin to become increasingly delicate, less resilient to mechanical stress, and more prone to damage. Combined with other age-related factors, this significantly raises the risk of chronic wounds in older adults [28]. Since stem cells play a key role in maintaining and repairing tissues, preserving both the cells and their surrounding environment is essential for effective anti-aging strategies.

*Andrographis paniculata (Burm. f.) Wall. ex Nees*, an annual herbaceous plant of the Acanthaceae family, is widely distributed throughout the South-East Asia region. The plant is known for its anti-inflammatory, antioxidant, and immune-boosting properties [29,30]. Andrographolide, the main active compound from *Andrographis paniculata*, has been reported to have beneficial effects in the treatment of inflammatory diseases [31], making it an interesting candidate for anti-aging cosmetic applications. Our work aimed to study the effect of an *Andrographis paniculata* extract (APE) on skin cells to assess its capacity to preserve KSC features and niche integrity.

## 2. Materials and Methods

### 2.1. Andrographis paniculata Extract (APE) Preparation and Analysis

**Plant Material and Extract Preparation:** *Andrographis paniculata* (Acanthaceae) plant material was sourced from China. The whole plants were mechanically processed through fine grinding and homogenization prior to extraction. An aqueous extraction was performed at 80 °C for 3 h, followed by filtration and purification using activated charcoal adsorption. The purified extract was subsequently solubilized in a vehicle consisting of butylene glycol and water (70:30, *v*/*v*) to achieve a final dry extract concentration of 1–10% (*w*/*w*). The resulting preparation appeared as a translucent liquid with pale yellow coloration. This optimized extraction and purification protocol was specifically designed to yield a standardized content of andrographolide, a bioactive labdane diterpenoid, ranging from 1000 to 25,000 ppm in the final ingredient.

**Analytical Characterization of Andrographolide Content:** To precisely quantify the andrographolide concentration and confirm extract composition, comprehensive chromatographic and spectroscopic analyses were conducted using liquid chromatography coupled with photodiode array detection and mass spectrometry (LC-PDA-MS).

**Instrumentation and Chromatographic Conditions:** The LC system consisted of an LC-30AD pump, SIL-30AC autosampler, CTO-20AC column oven, SPD-M30A photodiode array detector, and LCMS2020 single quadrupole mass spectrometer (all from Shimadzu, Kyoto, Japan). Chromatographic separation was achieved using an EVO C18 column (150 × 2.6 mm, 1.7 µm; Phenomenex, Torrance, CA, USA) maintained at 40 °C. The mobile phases comprised (A) 0.1% formic acid in water and (B) 0.1% formic acid in acetonitrile. The gradient elution program was as follows: 0–2 min, 10% B; 2–15 min, linear gradient from 10% to 60% B; 15–15.1 min, 60% to 100% B; 15.1–17 min, 100% B; 17–17.1 min, 100% to 10% B; 17.1–20 min, 10% B. The flow rate was maintained at 0.4 mL/min with an injection volume of 10 µL.

**Mass Spectrometry Conditions:** Mass spectrometric detection was performed in negative electrospray ionization mode with the following conditions: interface voltage −4.5 kV, interface temperature 350 °C, desolvation line temperature 250 °C, nebulizer gas flow 1.5 L/min, and heat block temperature 400 °C.

**Sample Preparation and Quantification:** Andrographolide standard stock solution (1 mg/mL) was prepared in methanol. Working standard solutions for calibration were prepared at concentrations of 10, 50, 100, 200, and 500 µg/mL. The plant extract samples were accurately weighed and diluted 5-fold with water in volumetric flasks. Compound identification was performed by LC-MS, while quantification was conducted using LC-UV at 235 nm. Data acquisition and processing were performed using LabSolutions software, version 5.93 (Shimadzu, Kyoto, Japan).

### 2.2. Human Keratinocyte Culture and Subpopulation Isolation

Human keratinocytes were isolated from pediatric foreskin epidermal waste material collected by the Azienda Ospedaliero-Universitaria Policlinico di Modena. All the samples were collected with written informed consent, in accordance with the Declaration of Helsinki, after approval of the Modena Medical Ethical Committee (Prot. 184/10). Fresh keratinocyte subpopulations, either enriched in keratinocyte stem cells (KSCs) or Transit amplifying (TA) cells, were separated based on their ability to adhere to type IV collagen (100 μg/mL; Sigma, St. Louis, MO, USA), as described in [32]. The cells were maintained in serum-free medium (KGM, Clonetics, San Diego, CA, USA) for further analysis at 37 °C, 5% CO_2_. The cells were derived from three independent donors.

### 2.3. Human Fibroblast Culture

Primary human dermal fibroblasts were isolated from pediatric foreskin waste material collected by the Azienda Ospedaliero-Universitaria Policlinico di Modena, recruited as specified above. Human fibroblasts (HFs) were obtained by explant culture and cultured in Dulbecco’s modified Eagle’s medium (DMEM, Euroclone S.p.A., Pero, MI, Italy) containing 5% fetal bovine serum (FBS, Sigma-Aldrich, St. Louis, MO, USA), 2 mM L-glutamine, and 100 μg/mL of penicillin–streptomycin, and maintained at 37 °C, 5% CO_2_. The cells were derived from three independent donors.

### 2.4. Human Skin Explant (HSE) Culture

Human skin tissues were obtained from surgical waste material in full respect of the Declaration of Helsinki and Article L.1243-4 of the French Public Health Code. The latter does not require any prior authorization by an ethics committee for sampling and using surgical residues. Normal skin explants were prepared from abdominal tissue derived from the surgery of a female donor (age 46 years, Caucasian, phototype III). The explants were stabilized in a proprietary medium at 37 °C, 5% CO_2_ for 24 h. Subsequently, APE at 1% in a cosmetic formulation or a placebo formulation (Water, Dicaprylyl Ether, Ammonium Acryloyldimethyltaurate/VP Crosspolymer, Citric Acid, Sodium Citrate, Phenoxyethanol, Methylparaben, Ethylparaben, Fragrance) was topically applied (2 mg/cm^2^) every day for 5 days. After 5 days, the culture was stopped, and the skin samples were bisected for two types of histological processing: paraffin embedding and cryopreservation. All the conditions were tested in triplicate.

### 2.5. Histo-Morphological Analysis of Skin Explants and Immunohistochemistry

Half of the skin samples were fixed in formalin, dehydrated, and embedded in paraffin. From these paraffinized blocks, 5 µm sections were obtained using a ThermoScientific™ HM340E microtome (Microm HM 340E, Fisher Scientific, Illkirch, France), dewaxed, rehydrated, and subjected to routine staining with the hematoxylin–Eosin–Saffron technique. Microscopic observations were performed using a Nikon Eclipse Ti2E microscope (Nikon Corporation, Tokyo, Japan), and pictures were digitized with a Nikon digital camera and the NIS-AR 5.42.04 software (Nikon Corporation, Tokyo, Japan). Epidermal thickness was evaluated using the Fiji Image2 software (version 2.16.0) (National Institutes of Health, Bethesda, MD, USA).

Immunohistochemistry (IHC) staining was performed using the IHC detection kit for rabbit/mouse primary antibody (#PK10006, Proteintech, Manchester, UK) according to the manufacturer’s instructions. The following markers were analyzed: Involucrin (mouse monoclonal #I9018, Sigma-Aldrich (Merck), Darmstadt, Germany), Claudin-1 (rabbit polyclonal #ab15099, Abcam, Cambridge, UK), Filaggrin (rabbit polyclonal #905801, BioLegend, san Diego, CA, USA), Ki-67 (rabbit polyclonal #ab15580, Abcam), and Survivin (rabbit polyclonal #ab76424, Abcam). Nuclei were counterstained with hematoxylin, as indicated in the figure legends. Images of immunohistochemistry staining were acquired using a D-Sight slide scanner (Menarini Diagnostics, Firenze, Italy). Quantification of positive area and epidermal nuclei count was performed using the Fiji ImageJ2 software (version 2.16.0). At least 3 different slides were analyzed. For each marker, the stained area was normalized to the number of epidermal nuclei. Then, the APE effect was calculated as a fold change versus placebo.

### 2.6. Fluorescent Immunolabeling of Skin Explants

Half of the skin samples were frozen at −80 °C, embedded in an optimal cutting temperature (OCT) embedding matrix (Tissue-Tek^®^ O.C.T.™, #4583, Sakura Finetech, Torrance, CA, USA), cut into 5 µm sections using a ThermoScientific™ HM550E Cryostat (Thermo Fisher Scientific Inc., Waltham, MA, USA), and stored at −20 °C until immunolabeling. The following markers were studied: Keratin 10 (rabbit polyclonal, #HPA012014, Atlas antibodies AB, Stockholm, Sweden), Collagen IV (rabbit polyclonal, #ab6586, Abcam), MMP9 (rabbit monoclonal, #ab76003, Abcam), total Laminin (rabbit polyclonal, #ab11575, Abcam), Collagen XVII (rabbit polyclonal, #HPA043673, Atlas antibodies), LRIG1 (mouse monoclonal, # MAB7498, R&D Systems, Minneapolis, MN, USA), Keratin 15 (mouse monoclonal, # ab80522, Abcam), and Integrin-β1 (rabbit monoclonal, # ab52971, Abcam). Briefly, after fixation of the skin sections in acetone, the slices were incubated with the primary antibody for two hours at room temperature (RT) then one hour with an Alexa Fluor^®^ 568 IgG anti-rabbit from goat (#A-11011, ThermoScientific™, Waltham, MA, USA) or with an Alexa Fluor^®^ 488 IgG anti-mouse from goat (#A-11029, ThermoScientific™) depending on the first antibody selected. Counterstaining of the nuclei was performed using Hoechst (#H3570, Invitrogen™,Carlsbad, CA, USA). The slides were mounted in aqueous medium, and microscopic observations were realized using a Nikon Eclipse Ti2E microscope (Nikon Corporation, Tokyo, Japan). Pictures were digitized with a Nikon digital camera and the NIS-AR software (Nikon Corporation, Tokyo, Japan). Quantification of the fluorescent area and epidermal nuclei count was performed using the Fiji ImageJ2 software (version 2.16.0). At least 3 different slides were analyzed. For each marker, the stained area was normalized to the number of epidermal nuclei. The APE effect was then calculated as a fold change versus placebo.

### 2.7. MTT Assay

Keratinocyte subpopulations and human fibroblasts were seeded in a 96-well tissue culture plate and treated with APE at different concentrations. An MTT (3-(4,5-dimethylthiazol-2-yl)-2,5-diphenyltetrazolium bromide, Sigma, St. Louis, MO, USA) assay was performed 48 h after treatment.

### 2.8. Colony Forming Efficiency (CFE)

Isolated keratinocyte subpopulations were plated at a density of 100 cells/dish in a 6-well plate on mitomycin C-treated human fibroblasts (2.4 × 104/cm^2^ and cultured in DMEM and Ham’s F12 media, with slight modifications to the protocol described in [33]. Fourteen days later, dishes were fixed and stained with 4% formaldehyde/1% Rhodamine B. Colonies containing more than 10 cells were counted, and CFE was calculated. The number of colonies was expressed as a percentage of the number of cells plated.

### 2.9. Co-Culture Model of DEJ Damage

Normal human fibroblasts (HFs) were seeded in multiple 6-well plates and, after 24 h, were treated with 0.05% APE for 3 days. At the end of the incubation period, HFs were treated with Mitomycin C to prepare the feeder layer. The following day, the Mitomycin C-treated fibroblasts were irradiated with UVB at 20 mJ/cm^2^ to mimic dermal damage. Six hours after UVB treatment, the KSC- and TA-enriched populations were seeded at very low density and immediately treated with 0.05% APE. After 48 h, the stimulus was removed, and the cells were cultured in DMEM and Ham’s F12 media for an additional 12 days. Culture medium was collected on days 2, 7, and 14 for further analysis. A schematic representation of this method is presented in Figure 7A.

### 2.10. ELISA Assay

Collagen IV and total Laminin production were evaluated in the culture medium of the treated cells using the Human COL4 ELISA kit (#AEFI00400, AssayGenie, Dublin, Ireland) and the Human LN ELISA kit (HUFI00191, AssayGenie), respectively. The latter detects total Laminin, referring to the complete heterotrimeric αβγ molecule according to the manufacturer’s instructions (AssayGenie, Dublin, Ireland).

### 2.11. Statistical Analysis

All the data are expressed as mean ± standard deviation (SD). Differences between the 2 groups were statistically analyzed using either Student’s *t*-test or the Mann–Whitney test depending on the normality check. For comparisons involving more than 2 groups, ordinary one-way ANOVA or two-way ANOVA was used, followed by Dunnett’s multiple comparisons test. Statistical analyses were performed using GraphPad Prism version 10.4.2 (GraphPad Software Inc., La Jolla, CA, USA). The level of statistical significance was set at 5%, with *p*-value indicated as follows: *p* < 0.05 (*), *p* < 0.01 (**), *p* < 0.001 (***), and *p* < 0.0001 (****).

## 3. Results

### 3.1. Andrographolide Content in Andrographis paniculata Extract (APE)

To characterize the *Andrographis paniculata* extract (APE), we conducted targeted liquid chromatography analysis focusing on andrographolide, a diterpene lactone extensively documented for its diverse pharmacological properties, including anti-inflammatory, anti-obesity, anti-diabetic, and anti-cancer activities [34]. The chromatographic analysis revealed that the predominant compound, exhibiting maximum absorbance at 235 nm in the APE, displayed a retention time identical to that of the andrographolide reference standard, consistent with previous reports of this compound’s presence in *A. paniculata* (Figure 1A). The identity of andrographolide was further confirmed through the LC-MS analysis, which demonstrated that the molecular mass of the compound in the extract matched that of the authentic standard (Figure 1B). A quantitative analysis using the external standard method determined an andrographolide concentration of 1900 ppm in the final formulation.

**Figure 1 cells-14-01176-f001:**
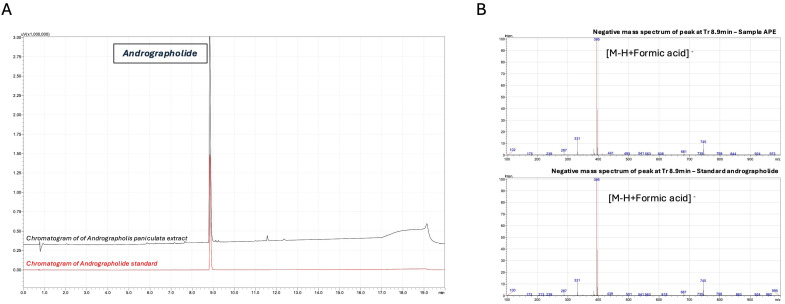
Chromatographic and mass spectrometric identification of andrographolide in *Andrographis paniculata* extract. (**A**) LC-UV chromatograms of *A. paniculata* extract (APE, black) and andrographolide reference standard (red) monitored at 235 nm. Separation was performed on an EVO C18 reverse-phase column (150 × 2.6 mm, 1.7 µm) using a gradient elution of 0.1% formic acid in water (A) and 0.1% formic acid in acetonitrile (B): 0–2 min, 10% B; 2–15 min, 10% to 60% B; 15–15.1 min, 60% to 100% B; 15.1–17 min, 100% B; 17–17.1 min, 100% to 10% B; 17.1–20 min, 10% B. Andrographolide was identified at retention time (tR) = 8.9 min. (**B**) Negative ion mode LC-ESI-MS spectra of APE (upper panel) and andrographolide reference standard (lower panel) at retention time 8.9 min. Analysis was performed using a single quadrupole mass spectrometer with an electrospray ionization source under the following conditions: interface voltage −4.5 kV, interface temperature 350 °C, desolvation line temperature 250 °C, nebulizer gas flow 1.5 L/min, and heat block temperature 400 °C. The predominant ion at m/z 395 corresponds to the deprotonated molecular ion [M-H+HCOOH]^−^ of andrographolide.

### 3.2. Improvement of Dermo-Epidermal Junction Composition by APE in Skin Explants

To assess the ability of APE to influence the behavior of epidermal stem cells, skin explants were treated daily with 1% APE for 5 days (Figure 2A). No alterations in tissue structure or epidermal thickness were observed, indicating good tissue tolerance. APE did not affect the expression of early (KRT10) or late (INV) differentiation markers, nor did it alter the expression of key barrier proteins Claudin-1 (CLDN1) and Filaggrin (FLG), suggesting that epidermal differentiation and barrier integrity were preserved (Figure 2B).

**Figure 2 cells-14-01176-f002:**
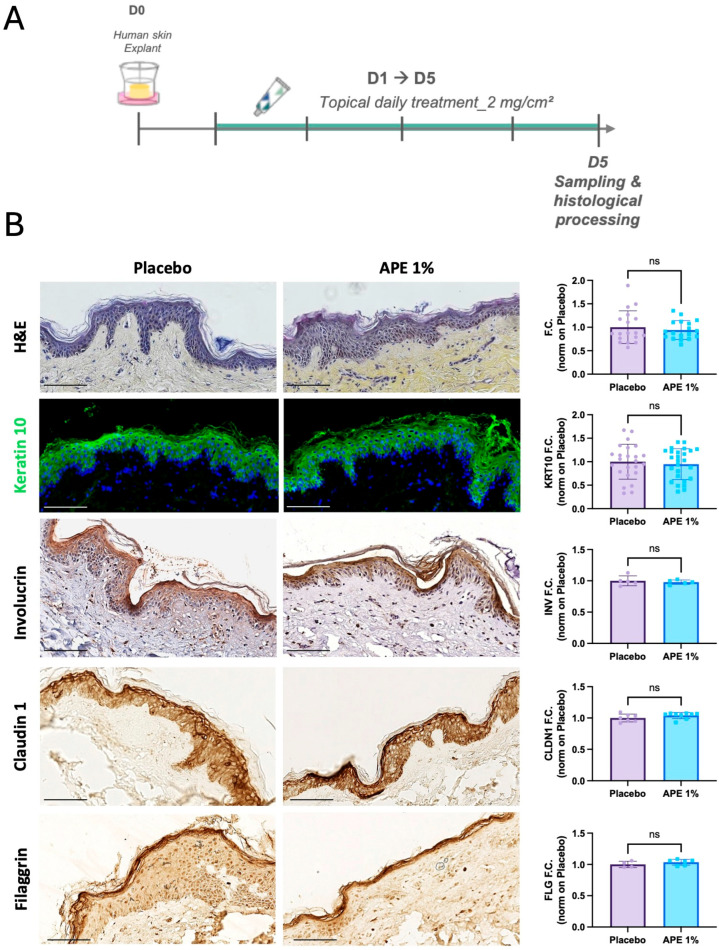
Skin morphology after daily applications of the APE formulation in ex vivo cultures. (**A**) After 5 days of daily topical application on skin explants of a formulation containing 1% of APE or placebo, the organ cultures were fixed and paraffin-embedded for histological and immuno-staining. (**B**) Sections were stained with H&E. Expression of early and late differentiation markers was evaluated by immunofluorescence (IF) and IHC staining, specifically KRT10 and INV, respectively. Nuclei were counterstained with Hoechst and hematoxylin, respectively. Expression of skin barrier markers, such as Claudin 1 (CLDN1) and Filaggrin (FLG) was evaluated by IHC staining. Quantification of skin thickness, and of KRT10, INV, CLDN1, and FLG expression levels was performed using the Fiji software. ‘ns’ indicates that the observed differences are not statistically significant (*p* > 0.05). Scale bar = 100 μm.

Importantly, APE treatment led to a significant improvement in the composition of the dermo-epidermal junction. Specifically, APE increased the expression of two key components of this structure: Collagen IV and Laminin. Collagen IV showed a statistically significant 2.58-fold increase compared to the placebo, while Laminin levels rose by 1.88-fold (Figure 3). Additionally, MMP-9, a matrix-degrading enzyme known to break down Collagen IV during aging, was downregulated by 0.41-fold (Figure 3). These findings suggest a strengthening of the dermo-epidermal junction, a critical skin structure impacted by aging, which plays a key role in the regenerative potential of the skin.

**Figure 3 cells-14-01176-f003:**
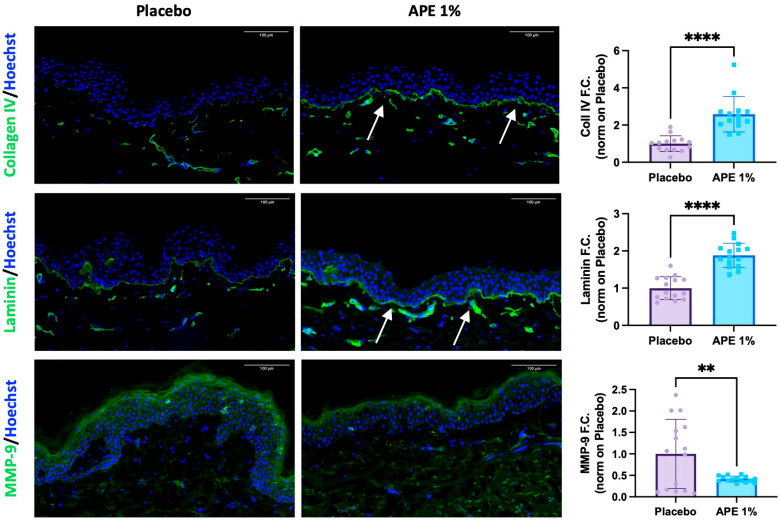
Increased DEJ key markers and protection against degradation after daily applications of APE formulation in ex vivo cultures. After 5 days of daily topical application on skin explants of a formulation containing 1% APE or placebo, the organ cultures were frozen and OCT-embedded. Expression of Collagen IV, total Laminin, and MMP-9 was evaluated by IF. Arrows highlight the oiserved increases. Quantification of the expression level was performed using the Fiji software. Statistical significance is indicated as follows: *p* < 0.01 (**), and *p* < 0.0001 (****). Counterstaining of the nucleus was performed using Hoechst. Scale bar = 100 μm.

### 3.3. Stemness Potential Preservation and Stimulation of Skin Cell Proliferative Capability by APE in Skin Explant

Markers of stemness potential were also investigated on the same skin explants. Protein expression of stem cell biology-related markers was increased 5 days after 1% APE application, highlighting its broad effect on the preservation of stem cell potential. One of the major niche components, Collagen XVII, was increased by 1.84, and β1-integrin, the receptor for cell anchorage within the niche, was increased by 1.34 (Figure 4A). Moreover, APE also regulated the proliferative capacity of the cells, as shown by the upregulation of Keratin 15 (KRT15) and Ki67 by 1.97- and 1.74-fold, respectively, and supported the maintenance of stem cell capacity and multipotency through the upregulation of Survivin (1.69-fold) and LRIG1 (2.5-fold) (Figure 4A,B). In detail, we observed increased nuclear expression of Survivin in the treated skin samples in contrast to the predominantly cytoplasmic localization seen in the placebo-treated samples (Figure 4B). Considering that nuclear Survivin is overexpressed in KSCs and plays a protective role against apoptosis [17], these results collectively suggest that APE supports a stem-like phenotype, as evidenced by the upregulation of markers associated with stem cell maintenance, proliferation, and niche integrity.

**Figure 4 cells-14-01176-f004:**
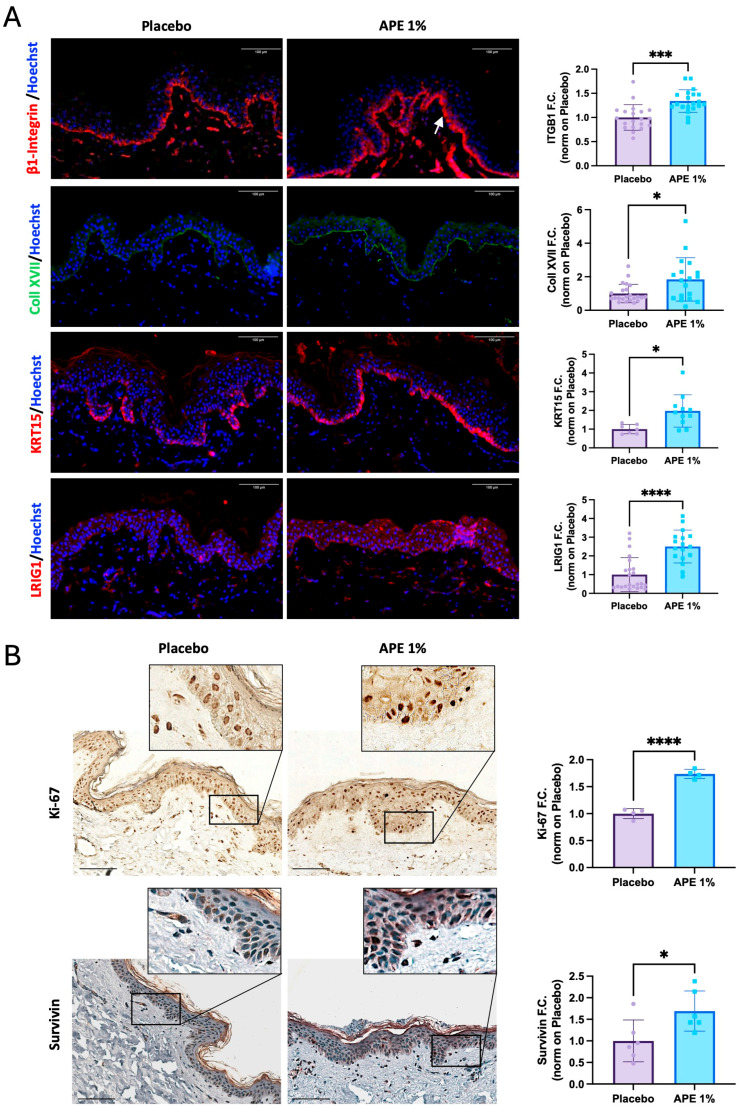
Increased stem cell and proliferative markers after daily applications of APE formulation in ex vivo cultures. After 5 days of daily topical application on skin explants of a formulation containing 1% APE or placebo, the organ cultures were fixed and frozen or paraffin-embedded. (**A**) Expression of β1-integrin (ITGB1), Collagen XVII (Coll XVII), Keratin 15 (KRT15), and LRIG1 was evaluated by IF. Arrow highlights the observed increase. Quantification of the expression level was performed using the Fiji software. Counterstaining of the nuclei was performed using Hoechst. (**B**) Expression of proliferative markers, Ki-67 and Survivin, was evaluated by IHC staining. Quantification of expression levels was performed using the Fiji software. Counterstaining of the nuclei in the Survivin samples was performed using hematoxylin. Statistical significance is indicated as follows: *p* < 0.05 (*), *p* < 0.001 (***), and *p* < 0.0001 (****). Scale bar = 100 μm.

### 3.4. Stimulation of Skin Cell Proliferative Capability by APE in Cell Cultures

We aimed to further investigate the effect of APE on the proliferative capacity of skin cells. We first sought to confirm the results obtained from skin explants in dispersed cultures of primary skin cells. The in vitro results supported that APE stimulates the proliferation of primary human fibroblasts, even at the lowest concentration tested (Figure 5A). To better characterize the cellular targets of APE, we performed MTT assays on subpopulations of primary keratinocytes, specifically stem cells (KSCs) and transit amplifying (TA) cells, isolated based on their differential adhesion to type IV collagen (Figure 5B). Our results indicate a significant proliferative response in the KSC population, while APE did not induce proliferation in human differentiated TA cells (Figure 5C), suggesting that APE selectively promotes the expansion of undifferentiated epidermal progenitors (Figure 5C). On the other hand, the clonogenic assay showed no differences in keratinocyte subpopulations treated with APE compared to the placebo, indicating that the stemness potential remains unaltered and epidermal homeostasis is preserved (Figure 5D).

**Figure 5 cells-14-01176-f005:**
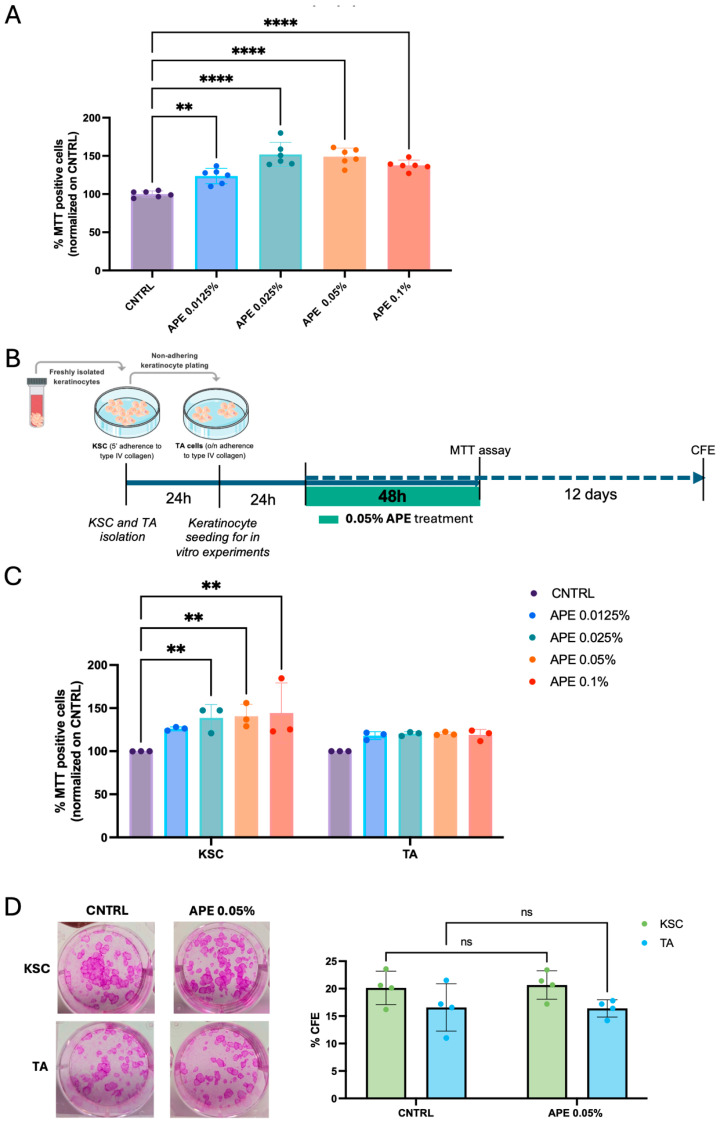
APE increased viability in KSC in vitro and preserved their clonogenic efficiency. (**A**) Human fibroblasts were seeded in a 96-well tissue culture plate and treated with APE at different concentrations. Cell proliferation was evaluated by MTT assay. The cells were derived from three independent donors. (**B**) Schematic representation of cell proliferation experiments with keratinocyte subpopulations. Keratinocytes were isolated based on their ability to adhere to type IV collagen: KSCs that adhere to the collagen in 5′, and TA cells that adhere overnight. After 24 h, the cells were reseeded in a 96-well tissue culture plate and treated with APE; 48 h later, an MTT assay was performed, and cell proliferation was analyzed. The cells were derived from three independent donors (**C**). (**D**) Moreover, KSC and TA cells were also reseeded in a 6-well tissue culture plate and treated with APE for 48 h. CFE assay was performed on culture day 14. Representative micrographs of CFE. Data represents the means of triplicate determinations. Statistical significance is indicated as follows: not statistically significant (ns: *p* > 0.05), *p* < 0.01 (**), and *p* < 0.0001 (****).

### 3.5. Stimulation of Skin Cell Laminin and Collagen Secretion by APE in Cell Cultures

Laminin and Collagen create a microenvironment that supports the self-renewal and differentiation of keratinocyte stem cells, ensuring proper skin homeostasis and repair. For this reason, we investigated whether APE modulates extracellular matrix composition, potentially influencing the maintenance and functionality of the stem cell niche. We measured the secretion of total Laminin and Collagen IV by keratinocyte subpopulations—specifically, KSCs and TA cells—following treatment with different doses of APE. Consistent with the observations in ex vivo skin samples, the levels of both Collagen IV and total Laminin, quantified by ELISA, showed a significant increase after 48 h of treatment (Figure 6). In particular, KSCs were more responsive to APE treatments, exhibiting elevated production of both Collagen IV and total Laminin at all the tested doses (Figure 6). Moreover, the increase in secretion appeared more pronounced for Laminin than for Collagen. In contrast, the TA cells showed a significant increase in Laminin secretion was observed only at the highest APE concentration (0.1%) (Figure 6). The differential responsiveness between the KSCs and TA cells suggests that APE may contribute to the reinforcement of the stem cell niche microenvironment, confirming that KSCs are a specific cellular target of APE.

**Figure 6 cells-14-01176-f006:**
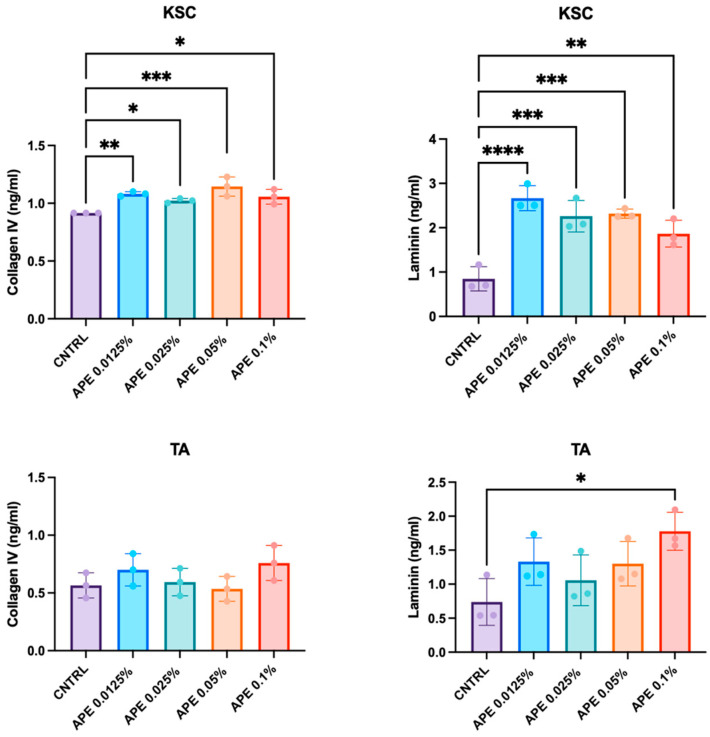
APE increased Collagen IV and total Laminin production in KSC in vitro. The KSCs and TA cells were treated with different doses of APE. Supernatants were collected 48 h later, and levels of Collagen IV and total Laminin, the complete heterotrimeric αβγ molecule, were evaluated by the ELISA analysis. Data represents the means of triplicate determinations. Statistical significance is indicated as follows: *p* < 0.05 (*), *p* < 0.01 (**), *p* < 0.001 (***), and *p* < 0.0001 (****).

### 3.6. APE Treatment Preserves KSC Capacity in Laminin Production in a DEJ Damaged Co-Culture Model

As solar UV exposure is a major causative factor in age-related changes [35], we aimed to assess whether APE can influence the behavior of epidermal stem cells also under UVB-induced damage. To this end, we first developed a model mimicking damage to the DEJ (Figure 7A). First, the human fibroblasts were treated with 0.05% APE for 72 h and subsequently with Mitomycin-C to prepare a classic feeder layer, non-dividing cells that provide extracellular secretions to support keratinocyte proliferation. The feeder layer was then subjected to UVB irradiation to simulate dermal damage, while sham-irradiated fibroblasts served as control (CNTRL). Following UVB exposure at 20 mJ/cm^2^, human keratinocyte stem cells (KSCs) or transit amplifying cells (TA) were seeded onto the feeder layer and treated with 0.05% APE for 48 h. Culture medium was collected at three time points: 48 h post-seeding (day 2), day 7, and day 14 (Figure 7A). The time-course production of Collagen IV and Laminin by the KSC and TA cells was quantified under both the control and UVB-stressed conditions. The results showed a time-dependent increment in the levels of these extracellular matrix proteins at different time points (day 2, day 7, and day 14), which was more pronounced in the KSC population. UVB exposure completely abrogated Collagen IV production over time (Figure 7B), while Laminin production was broadly reduced by UVB but remained detectable and noteworthy (Figure 7B). Despite the UVB-induced DEJ damage, the APE treatment did not affect keratinocyte morphology or induce signs of cellular distress (Figure 7C). Interestingly, an increase in total Laminin production was observed exclusively in the APE-treated KSCs at day 2 (Figure 7D); however, this effect was transient and no longer detectable by day 14, when the Laminin levels returned to those of the control group. This response was specific to the KSC population, as TA cells did not exhibit a comparable increase in Laminin production (Figure 7D), confirming a differential sensitivity to APE. These findings indicate that APE may exert a protective or restorative effect specifically on KSC co-cultured on a UVB-stressed feeder layer, highlighting its potential as an agent to mitigate UVB-induced damage and promote extracellular matrix stability.

**Figure 7 cells-14-01176-f007:**
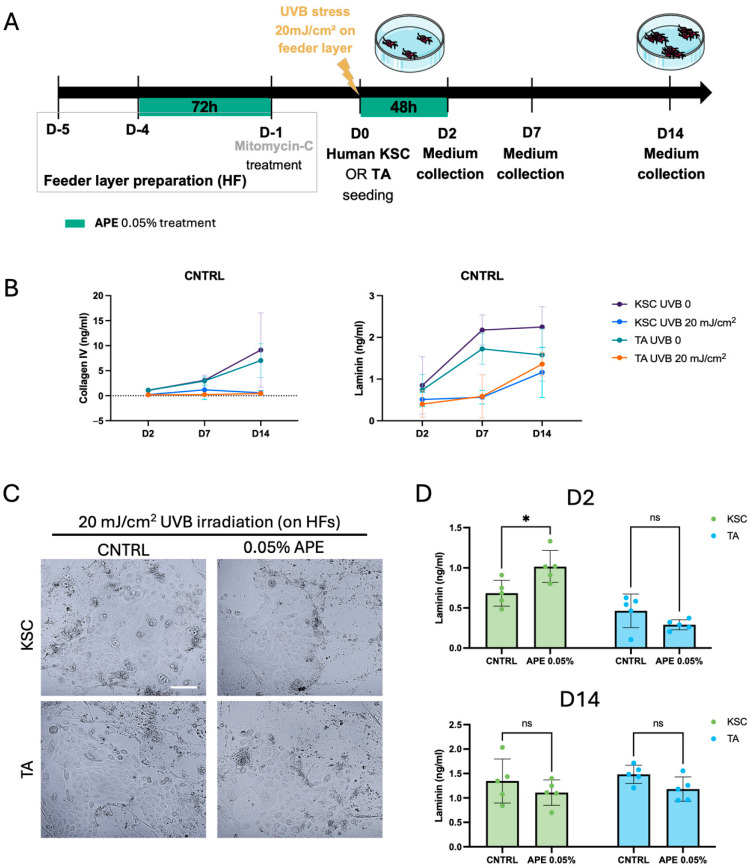
APE increased the Laminin production of KSC on damaged DEJ in a co-culture model. (**A**) Schematic representation of the experiment in the co-culture model of the KSC and TA cells with the UVB-damaged human fibroblast feeder layer. The sham-irradiated (UVB 0) fibroblasts served as control (CNTRL). (**B**) Supernatants were collected at 2, 7, and 14 days after seeding on fibroblasts feeder layer with or w/o UVB treatment. Levels of Collagen IV and total Laminin, the complete heterotrimeric αβγ molecule, were evaluated by the ELISA analysis. (**C**) Micrographs of cells in co-culture with the UVB-treated HFs. Scale bar = 100 μm (**D**) Supernatants of the UVB-treated co-culture were also collected at day 2 (D2) and day 14 (D14), after the APE treatment. The Laminin levels were evaluated by the ELISA analysis and compared to CNTRL. Data represents the means of quintuplicate determinations. Statistical significance is indicated as follows: not statistically significant (ns: *p* > 0.05), and *p* < 0.05 (*).

## 4. Discussion

The dermo-epidermal junction (DEJ) plays a critical role in maintaining skin integrity and function, acting as a structural support for the epidermis and facilitating communication between the dermis and epidermis. Aging and environmental stressors, such as UV radiation, can compromise the DEJ, leading to diminished skin barrier function and reduced stem cell potential [36]. Recent studies have highlighted the importance of natural extracts in skin health, with *Andrographis paniculata* extract (APE) emerging as a promising candidate due to its antioxidant and anti-inflammatory properties [37,38].

*Andrographis paniculata* has a well-established ethnopharmacological history, with traditional uses spanning multiple therapeutic domains, including anti-inflammatory, anti-obesity, anti-diabetic, and anti-cancer activities. A comprehensive systematic review and meta-analysis have substantiated its favorable safety profile for oral administration, supporting its use in traditional medicine and dietary supplements [39]. While dermatological applications remain less extensively documented in the scientific literature, emerging evidence suggests promising cutaneous benefits. Recent research demonstrated that specific diterpenoids from *A. paniculata* target cutaneous TRPV3 channels, effectively alleviating pruritus [40]. Andrographolide, the principal bioactive constituent of *A. paniculata* extract, has been extensively characterized for its diverse pharmacological activities. Recent investigations have shown potential anti-psoriatic effects of andrographolide in comparative studies with topical corticosteroids, suggesting its therapeutic value in psoriasis management [41]. The development of a standardized *A. paniculata* extract with optimized andrographolide content represents a strategic approach to harnessing these benefits for dermatological applications aimed at improving skin quality and function.

The results of this study demonstrate the potential of *Andrographis paniculata* extract (APE) in enhancing the composition of the dermo-epidermal junction (DEJ) and preserving the stemness potential of keratinocyte stem cells (KSCs) in skin explants and in cell cultures. Our findings on skin explants indicate that daily treatment with 1% APE significantly enhances the composition of the DEJ by increasing the expression of Collagen IV and Laminin while downregulating MMP-9, a matrix-degrading enzyme [27]. Importantly, APE does not alter the tissue structure or organization, nor does it affect the epidermal thickness or skin barrier markers such as CLDN1 and FLG. Additionally, APE does not modulate the expression of early and late differentiation markers, including KRT10 and INV, respectively. Altogether, APE appears to strengthen the DEJ, a key structure for supporting the interfollicular epidermal stem cell niche and maintaining stemness potential of the skin [42,43] while being well tolerated and not interfering with keratinocyte differentiation or epidermal barrier function.

The preservation of stemness potential in epidermal cells is essential for maintaining skin homeostasis and regenerative capacity. Keratinocyte stem cells play a central role in skin repair and regeneration, supported by a niche microenvironment composed of various extracellular matrix components and signaling molecules [42]. *Andrographis paniculata* has been studied for its beneficial effects on skin health, including its ability to enhance the stem cell microenvironment and promote cell proliferation [29,30]. Given the lack of a single definitive marker for interfollicular epidermal stem cells, we demonstrated that APE treatment positively influences the stemness potential of epidermal cells by modulating the expression of multiple stem cell-associated markers commonly used to identify stem-like populations. Protein expression of niche markers, including Collagen XVII and β1-integrin, was increased, highlighting APE’s role in preserving the stem cell microenvironment. Furthermore, APE enhanced the proliferative capacity of skin cells through the upregulation of KRT15 and Ki-67, and supported stem cell maintenance and multipotency via increased expression of Survivin and LRIG1. Notably, the increased nuclear expression of Survivin in the treated samples suggests a protective role against apoptosis [17], collectively indicating that APE supports a stem-like phenotype, as evidenced by the upregulation of the markers associated with stem cell maintenance, proliferation, and niche integrity. In vitro studies confirmed that APE stimulates both primary human fibroblast and KSC populations, while having no proliferative effect on differentiated TA cells, suggesting selective promotion of undifferentiated epidermal progenitors. The clonogenic assays further indicated that APE preserves stemness potential without altering epidermal homeostasis. Moreover, APE modulates extracellular matrix composition by increasing the secretion of Laminin and Collagen IV in keratinocyte subpopulations, particularly in KSCs, reinforcing the stem cell niche microenvironment and confirming KSCs as a specific cellular target of APE.

Nonetheless, we acknowledge that the identification of interfollicular epidermal stem cells remains a complex and evolving area of research. To date, no single marker or universally accepted panel exists for their definitive identification in human skin. For this reason, we adopted a combinatorial approach, selecting markers that are widely recognized in the literature to be associated with stemness, proliferation, and niche components. However, our conclusions are supported by both marker expression and functional assays performed on the KSC- and TA-enriched populations, which confirmed the selective proliferative effect of APE on undifferentiated progenitors. These complementary approaches reinforce the interpretation that APE contributes to the maintenance of a stem-like phenotype and supports the integrity of the stem cell niche. Future studies will aim to expand the marker panel and include transcriptional profiling to further refine the characterization of stem cell populations and their response to APE treatment.

Exposure to UVB radiation is known to cause significant damage to the skin, including DNA damage, oxidative stress, and disruption of extracellular matrix components [44]. Indeed, in our model of UVB-induced damage to the DEJ, we observed a dramatic downregulation of Collagen IV and Laminin production. These findings are in line with previous studies demonstrating that photoaging leads to a decrease in laminin-511 in the basement membrane, which in turn reduces the population of epidermal stem/progenitor cells [19]. KSCs play a crucial role in maintaining skin homeostasis and repair, and their ability to withstand UVB-induced damage is vital for effective skin regeneration [45,46]. Notara et al. found that short-term UVB irradiation can cause damage to limbal stem cells and upregulate macrophage-recruiting cytokines, further highlighting the detrimental effects of UV radiation on stem cell niches [47]. APE has been shown to exert a protective or restorative effect on KSCs co-cultured on a feeder layer exposed to UVB stress. The increase in Laminin production by KSCs under UVB stimulation indicates that APE may mitigate UVB-induced damage and promote extracellular matrix stability. These findings are consistent with previous research [29,30], which explored the active constituents of *Andrographis paniculata* in protecting the skin barrier and their synergistic effects with Collagen XVII. It has been demonstrated that *Andrographis paniculata* enhances skin barrier function and works synergistically with Collagen XVII to support skin health and repair mechanisms.

## 5. Conclusions

This study demonstrates that *Andrographis paniculata* extract (APE), standardized for andrographolide content, exerts protective and regenerative effects on the dermo-epidermal junction and KSCs. APE enhances the composition of the DEJ by increasing Collagen IV and Laminin, reducing MMP-9, and promoting the proliferative capacity of skin cells without altering tissue homeostasis or differentiation. APE selectively stimulates undifferentiated KSCs, as confirmed by functional assays, and shows protective effects in a UVB-induced damage model. These findings highlight the potential of APE to reinforce the stem cell niche.

Although no definitive marker exists for interfollicular epidermal stem cells, our combinatorial approach and functional validation provide a solid framework. Future studies will explore the underlying signaling pathways and assess the long-term efficacy in clinical settings.

Overall, APE emerges as a promising bioactive compound for dermo-cosmetic applications aimed at restoring skin architecture and enhancing regenerative capacity.

## Data Availability

Data supporting the findings of this study are available from the corresponding authors upon reasonable request.

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
