# Peer review of "Andrographis paniculata Extract Supports Skin Homeostasis by Enhancing Epidermal Stem Cell Function and Reinforcing Their Extracellular Niche"

_cells, 2025, doi:10.3390/cells14151176_

Round 1
Reviewer 1 Report
Comments and Suggestions for Authors
The manuscript lacks critical information and demonstrates several scientific inaccuracies and omissions that severely undermine its credibility and value. The following key issues must be addressed:
- Title of the Manuscript
Synergistic effects seem to me to be overstated as authors have no evidence showing that the effects on stem cells and the microenvironment are interdependent and greater than additive.
Revised title - Andrographis paniculata extract supports skin homeostasis by enhancing epidermal stem cell function and reinforcing their extracellular niche
2. Lack of Product Identity
The manuscript fails to disclose essential information regarding the identity and composition of the product(s) used in the study. Without such details, the study is not reproducible and lacks scientific rigor. Full disclosure of the product composition, identification of active ingredients, and excipients (if used) is necessary to validate and interpret the results.
- Preserves Stemness
Authors are claiming that APE “preserves stemness” based solely on the upregulation of a few general markers, it is indeed a bold and likely overstated conclusion, for several reasons. I am just giving a few examples
- Many plant extracts and/or plant derived compounds influence cell survival, oxidative stress, or proliferation.
- These effects can non-specifically increase marker expression (such as, Survivin) without truly affecting stemness.
- Markers are not definitive for stemness
- Ki-67: Proliferation; not specific to stem cells.
- Survivin: Anti-apoptotic; overexpressed in many cancer and non-stem cells.
- Collagen XVII, β1-integrin: Involved in adhesion; yes, found in stem cells but also elsewhere.
- LRIG1: Stronger marker, but needs to be shown alongside function (e.g., self-renewal capacity).
- Skin Homeostasis Claim
Skin homeostasis encompasses the dynamic balance required to maintain the structural and functional integrity of the skin. While this includes multiple elements, such as epidermal renewal, extracellular matrix stability, immune regulation, and responses to environmental stressors; barrier integrity remains the most fundamental component. Therefore, to substantiate a skin homeostasis claim for Andrographis paniculata extract (APE), it is essential to provide evidence that it contributes to the maintenance or improvement of skin barrier function, alongside its demonstrated effects on epidermal stem cells and the dermal-epidermal junction. The manuscript lacks this key information.
- Human Skin Explant Study
This study lacks reproducibility because only one tissue from an abdominal surgery of a female donor (age 46 years, Caucasian, phototype III) has been used (Section 2.4). This study must be replicated by using at least one more explant.
Recommendation: I recommend that this manuscript be rejected for publication in its current form. The authors must address all major concerns for the study to be considered credible and scientifically valuable.
Author Response
Please see the attached Q&A file

Reviewer 2 Report
Comments and Suggestions for Authors
The manuscript by Robert Lotti et al. deals with the potential of Andrographis paniculata extract (APE) as a component supporting skin homeostasis, with particular emphasis on its action on epidermal stem cells (KSC) and the structure of the dermal-epidermal junction (DEJ). The manuscript presents new knowledge in the field of regenerative dermatology, combining the biological mechanisms of action of the plant extract with modern ex vivo and in vitro research models. The great advantage of the article is the use of different research techniques. The results are presented clearly and thoroughly discussed. The experiments are planned correctly. The literature is cited correctly. The article presents a high scientific value and provides a solid basis for further research on Andrographis paniculata extract as a component of anti-aging products.
As a minor suggestion, I suggest including information on potential side effects of Andrographis paniculata extract in the manuscript.
Reviewer 3 Report
Comments and Suggestions for Authors
Lotti et al. report that Andrographis paniculata extract (APE) affects human epidermal stem cells and “promotes skin homeostasis”. Although paper is interesting, it requires substantial revision.
The title does not contain a clear message. As “skin homeostasis” cannot be measured, it is unclear what promotion means here.
Several data do not demonstate convincing differences due to APE. The legend of Figure 6 claims that APE increased the Laminin production of KSC on damaged DEJ in a co-culture model. Most of the data are not marked as significant in this figure.
Add more details in the section “Andrographis paniculata extract (APE) preparation”. As the method is presented now, the experiment cannot be repeated. The description is not scientifically valid. Mention the source of the plant.
Provide more information about Andrographis paniculata. The description “Andrographis paniculata is a natural plant” is not informative. Does it have a common name? To which taxonomic family does it belong?
The Conflicts of Interest must be changed. The authors declare no conflicts of interest. This is not possible because several authors are employed by companies with a commercial interest in this study.
Reviewer 4 Report
Comments and Suggestions for Authors
The manuscript entitled " Andrographis paniculata extract promotes skin homeostasis through synergistic effects on human epidermal stem cells and their microenvironment “ is informative, but there are some concerns that need to be addressed as follows.
Major Concerns
The data presented are interesting but still not adequate observation using crude extract of Andrographis paniculata whole plants (APE). Therefore, the manuscript needs more solid evidence based on the detailed chemical characterization of APE.
The author has shown that Andrographis paniculata extract significantly enhances the composition of the dermo-epidermal junction (DEJ) and preserves the stemness potential of keratinocyte stem cells (KSCs) against UVB-induced damage.
Recent studies in diverse organs, however, have highlighted dedifferentiation of specified tissues (skin or intestinal lining) as the dominant means for tissue regeneration (Ref. 1).
Regarding this point, the author should confirm whether the effects of APE on KSCs is due to the recovery from injury by tissue dedifferentiation.
Ref. 1
Tissue regeneration: Reserve or reverse?
Shivdasani RA, Clevers H, de Sauvage FJ.Science. 2021 Feb 19;371(6531):784-786.
Round 2
Reviewer 1 Report
Comments and Suggestions for Authors
My conclusion for this manuscript has not changed and I am keeping “rejection’ status intact for the following reasons. Some work is needed to improve English.
Abstract
Authors states – “In vitro studies revealed that APE selectively stimulated KSC proliferation without affecting transit amplifying cells, and enhanced Collagen IV and Laminin secretion, particularly in KSCs. Furthermore, in a co-culture model mimicking a compromised DEJ (UVB-induced), APE increased Laminin production in KSCs, suggesting a protective effect against photo-aging. These findings indicate that APE enhances DEJ composition.”
Above statement can be made, however, I have problem with the use of terminologies, like “selectively” and “photo-aging” — they should be well-supported by the data (e.g., specific proliferation assays for KSCs vs TACs, and aging markers). This information is lacking.
Product identity
Surprisingly, authors have used water as a solvent for obtaining andrographolide which has reported water-solubility of only about 3.29 µg/mL (or about 0.00329 mg/mL) at room temperature. Can authors explain this selection?
Preserving stemness
The following pattern of marker expression is consistent with the preservation of a stem-like state.
- Niche anchorage (Collagen XVII, β1-integrin) – important for maintaining the stem cell microenvironment.
- Proliferation (Ki-67, KRT15) – indicative of cell cycle activity and basal epithelial cells.
- Stemness (Survivin, LRIG1) – linked to stem cell maintenance and self-renewal.
The conclusion is based only on marker expressions. Therefore, a statement like this is more appropriate – “APE supports a stem-like phenotype, as evidenced by the upregulation of markers associated with stem cell maintenance, proliferation, and niche anchorage.”
However, authors should have focused on selecting a core panel for human epidermal stem cells, a practical marker set might include ITGA6^high/CD71^low, KRT14+/KRT10–, and ΔNp63+ with or without Ki-67, depending on stem cell activation status. At a minimum, TP63 should be included, as this transcription factor is considered a master regulator of epidermal stemness and is essential for maintaining self-renewal. This information is missing.
Skin Homeostasis Claim – Authors have not provided any response
Skin homeostasis encompasses the dynamic balance required to maintain the structural and functional integrity of the skin. While this includes multiple elements, such as epidermal renewal, extracellular matrix stability, immune regulation, and responses to environmental stressors; barrier integrity remains the most fundamental component. Therefore, to substantiate a skin homeostasis claim for Andrographis paniculata extract (APE), it is essential to provide evidence that it contributes to the maintenance or improvement of skin barrier function, alongside its demonstrated effects on epidermal stem cells and the dermal-epidermal junction. The manuscript lacks this key information.
Results: Section-3.2
Authors states that “this could suggest that APE did not affect the process of keratinocyte differentiation or the formation of the epidermal barrier.” There is no need to show negative results by elaborating with figures and contents. This should be stated in a few sentences, not two pages of negative data and information. These results invalidate the statement above.
Comments on the Quality of English LanguageSames as above
Reviewer 3 Report
Comments and Suggestions for Authors
Thank you for addressing the comments.
Author Response
We are grateful to read that the reviewer is satisfied by the changes apported to the manuscript. We thank the reviewer for her/his thorough review.
Reviewer 4 Report
Comments and Suggestions for Authors
The manuscript has been well improved.
Author Response
We thank the reviewer for accepting our manuscript and we are thankful for her/his friendly and supportive analysis of our study.